# Retroviral RNA Processing

**DOI:** 10.3390/v14051113

**Published:** 2022-05-23

**Authors:** Karen L. Beemon

**Affiliations:** Biology Department, Johns Hopkins University, Baltimore, MD 21218, USA; klb@jhu.edu

**Keywords:** retroviruses, foamy viruses, RNA processing, export, splicing, translation, RNA modification, NMD

## Abstract

This review is an accompaniment to a Special Issue on “Retroviral RNA Processing”. It discusses post-transcriptional regulation of retroviruses, ranging from the ancient foamy viruses to more modern viruses, such as HIV-1, HTLV-1, Rous sarcoma virus, murine leukemia virus, mouse mammary tumor virus, and Mason-Pfizer monkey virus. This review is not comprehensive. However, it tries to address some of the major questions in the field with examples of how different retroviruses express their genes. It is amazing that a single primary RNA transcript can have so many possible fates: genomic RNA, unspliced mRNA, and up to 50 different alternatively spliced mRNAs. This review will discuss the sorting of RNAs for packaging or translation, RNA nuclear export mechanisms, splicing, translation, RNA modifications, and avoidance of nonsense-mediated RNA decay.

## 1. Introduction

This Special Issue of *Viruses* is devoted to **Retroviral RNA Processing**. We will focus on new developments in this field and will include representatives of several types of retroviruses. Topics in this Special Issue include regulation of HIV-1 alternative splicing during viral replication [1] and latency/persistence [2]. In addition, tri-methyl G (TMG) capped HIV-1 RNA is reviewed [3]. New live cell imaging methods are developed to study Rev-driven export of HIV RNAs from the nucleus [4]. Imaging is also used to show cotranscriptional dimerization of Rous Sarcoma Virus (RSV) RNA [5]. In addition, the role of R loops and genomic instability in HTLV-1-infected cells will be reviewed [6]. Unfortunately, four potential contributors canceled their submissions, due in part to complications of the COVID-19 pandemic. Earlier reviews on this topic include [7,8,9,10].

## 2. Background

One of many unusual features of orthoretroviruses is that they package two identical copies of their approximately 10 kb, single-stranded RNA genome; thus, the RNA genome in virions is diploid [11,12,13,14]. Unlike most plus-stranded RNA viruses, retroviruses do not translate their genomic RNA immediately after entering the cell’s cytoplasm. Instead, the retroviral RNA genome is contained within the capsid, along with the viral-encoded enzymes: reverse transcriptase and integrase. The reverse transcriptase makes a single, double-stranded DNA copy of the RNA genome within the capsid after entry into the cell. This is transported to (or partially synthesized in) the nucleus and integrated into the host DNA by the viral integrase [7]. This integration process allows the viral DNA genome to be stably maintained as part of the host genome. It also allows it to be transcribed and processed by host machinery. Integrated viral DNA is transcribed at a much higher rate than unintegrated DNA. The unintegrated DNA is transcriptionally silenced by associated histones [15]. If two different viruses infect the same cell, heterodimers can be packaged, allowing recombination during reverse transcription [11].

Cleverly, the viral cDNA reverse transcript generates transcriptional regulatory sequences at each end of the genome in its Long Terminal Repeat (LTR) [7]. The LTR includes transcriptional enhancers and promoters, as well as polyadenylation regulatory sequences. Transcription by host RNA Polymerase II generates a single primary RNA transcript that is identical to the packaged genomic RNAs (give or take a nucleotide or two at the 5′ end in some cases). This RNA has a 5′ cap and 3′ poly(A) sequence, like cellular mRNAs. In addition, it can be modified internally by host enzymes to generate m^6^A and m^5^C residues.

Retroviral gene expression uses a “hybrid” model, involving both unspliced and spliced mRNAs. The unspliced full-length primary transcript is the mRNA that generates Gag and Gag-Pol polyproteins. This primary transcript is also the genomic RNA that is packaged into viral particles as dimers. The Gag and Gag-Pol polyproteins are subsequently cleaved by the viral-encoded protease after budding from the cell. The Gag polyprotein is processed to generate the capsid proteins: matrix, capsid, nucleocapsid, and additional, virus-specific small Gag proteins. The retroviral enzymes are generated from the Pol polyprotein and comprise protease, reverse transcriptase, and integrase.

The gene(s) at the 3′ end of the genome is expressed from subgenomic mRNAs, generated by splicing of the primary RNA transcript. All retroviruses generate a singly spliced *env* mRNA, encoding the viral glycoproteins on the surface of the virion. Other singly spliced mRNAs include RSV *src*, HIV *vpu* and *vif*, and PFV *pol*. In addition, multiply spliced mRNAs encode accessory proteins, including HIV-1 Tat and Rev. However, a large fraction of the primary transcripts must remain unspliced for genomic RNA and Gag–Pol mRNAs, so splicing needs to be tightly regulated [1]. The complex retroviruses express accessory proteins from alternatively spliced mRNAs. HIV-1 has over 50 different mRNAs, all derived from the same full-length pre-mRNA [1,16].

This hybrid model generates several interesting RNA processing problems that retroviruses need to overcome: (1) Primary RNA transcripts must be sorted into packaged and translated RNAs, although they are thought to be essentially identical in sequence. (2) The translated mRNAs must be sorted into unspliced and spliced fractions, and the alternative splicing must be regulated to generate numerous products in the complex viruses. (3) The unspliced and incompletely spliced intron-containing RNAs must be exported to the cytoplasm, contrary to the normal rules of eukaryotic mRNA processing. This is necessary because introns perform a double duty as coding sequences. (4) Different amounts of *gag* and *pol* gene products need to be synthesized from the same mRNA. (5) Since the unspliced mRNA has a long 3′ UTR, it is a target for nonsense-mediated RNA decay (NMD). This must be inhibited to retain stable full-length mRNA for viral replication.

This review will discuss numerous examples where different retroviruses use different methods to solve these problems. This Special Issue on Retroviral RNA Processing focuses on research investigating how retroviruses usurp cellular machinery to process their RNAs and express their genes.

## 3. Foamy Virus RNA Processing

Foamy viruses are the most ancient retroviruses and comprise the subfamily Spumaretroviruses. Endogenous spumaviruses date back to the first jawed vertebrates (Coelecanths) over 450 million years ago [17,18]. They are nonpathogenic in a host of vertebrates, including nonhuman primates. They can infect homo sapiens but are not transmitted from person to person [19].

The spumaviruses’ mode of RNA processing has some interesting differences from most orthoretroviruses (reviewed in [19]). Like all retroviruses, they encode *gag*, *pol*, and *env* genes. In addition, genes for two accessory proteins, Tas and Bet, are encoded at the 3′ end of the genome, making these complex retroviruses. Unusually, an internal promoter within the *env* gene is expressed early after infection to generate the accessory proteins from spliced mRNAs. Tas is a transcriptional activator that binds DNA, both in the LTR and at the internal promoter. It is expressed early after infection, and its expression leads to a switch from use of the internal promoter to the 5′ LTR promoter. This induces the expression of the viral structural genes later in infection. The Bet protein is an antagonist of APOBEC3 and prevents packaging of this deaminase [20]. It is interesting that no homolog of Rev has been detected in the foamy virus genome.

The full-length foamy virus RNA transcript functions as the mRNA for the N-terminal Gag protein. In contrast to the orthoretroviruses, Pol is expressed from a singly spliced mRNA and not as a Gag–Pol fusion polyprotein [21]. Foamy virus Env is expressed from a singly spliced mRNA like other retroviruses. There is less proteolytic processing of the foamy virus Gag and Pol proteins than with the orthoretroviruses. Gag is only processed once to remove a 3K fragment from its C-terminus. The Pol polyprotein is also cleaved only once between integrase and reverse transcriptase. Thus, protease and reverse transcriptase are expressed as a fusion protein. Orthoretroviruses package Pol as a Gag–Pol fusion protein, and its packaging is promoted by Gag–Gag interactions. In contrast, foamy viruses are thought to package Pol by interactions with the Gag protein and/or the viral RNA [22].

Nucleocytoplasmic export of the foamy unspliced RNA and spliced (but intron-containing) *pol* and *env* mRNAs occurs via the CRM1 pathway, but its use is novel because there is no homolog of the HIV-1 Rev regulatory protein. The cellular shuttle protein HuR binds the foamy viral RNA and interacts with CRM1 via the bridging proteins ANP32A and ANP32B [23]. HuR also promotes RNA translation and stability. Tas and Bet accessory proteins are synthesized from fully spliced mRNAs early after infection, and their expression is not CRM1-dependent.

In major contrast to orthoretroviruses, reverse transcription of the packaged RNA dimer begins in the cytoplasm before it exits the cell and is thought to be completed before the virus infects a new cell [19]. In this way, the foamy viruses are similar to hepadnaviruses such as the Hepatitis B virus. However, the foamy DNA genome is linear, while the hepadna genome is a partially double-stranded circular DNA. Thus, the foamy virion contains DNA and possibly also RNA. After infection, its DNA is integrated into the host genome, like other retroviruses.

The Betaretrovirus mouse mammary tumor virus (MMTV), like foamy viruses, has an internal promoter and splice donor (SD) in *env* (reviewed in [24]). In MMTV, the internal promoter drives expression of the Sag (super antigen) gene from the large LTR. MMTV virions assemble in a cytoplasmic compartment before transport to the plasma membrane for budding. Unlike the foamy viruses, MMTV has a Rem protein used for export that interacts with a RemRE at the *env*/U3 junction [25].

## 4. Viral RNA Must Choose between Packaging and Translation (or Splicing)

After integration into the host genome, proviruses are transcribed by host RNA polymerase II. Unspliced primary RNA transcripts can escape splicing and be exported from the nucleus to become packaged genomic RNA or translated mRNA. How and where is this selection between packaging and translation made? Alternative RNA structures in the 5′ UTR have been proposed to distinguish genomic from translated RNAs [26,27,28,29].

Interestingly, a small difference in the sequence of the 5′ end of HIV-1 RNA has been observed between packaged genomic RNA and translated mRNA. The packaged RNA has a single transcribed G residue at its 5′ end, while the mRNAs have two or three transcribed G residues [30,31,32]. In addition, all have a cap that is added post-transcriptionally. This small change in the transcription initiation site generates quite different RNA secondary structures at the 5′ end of the full-length viral RNA, probably cotranscriptionally. The packaged RNA structure favors RNA dimerization through a kissing loop dimer initiation sequence (DIS) and exposed psi packaging sequence and sequesters the cap and the *gag* AUG translation initiation codon. In contrast, the translated mRNA favors translation with an exposed cap and AUG, while the DIS is sequestered. The presence of a TMG cap on HIV RNA [3] may further affect the structure of the 5′ UTR.

Differences in transcription initiation sites and the presence of TMG caps have not been observed for retroviruses other than HIV-1. However, the TMG cap appears to be associated with CRM1-mediated export, so may be found in other complex retroviruses. Other factors that might help distinguish between RNA destined for packaging or translation include RNA dimerization, RNA modification, and binding to Gag or cellular splicing factors or ribosomes.

## 5. Is RNA Dimerization Cotranscriptional?

HIV-1 RNAs appear to be exported to the cytoplasm as monomers and to dimerize during virion assembly at the plasma membrane [33,34]. In contrast, some murine leukemia virus (MLV) full-length viral RNA constructs have been proposed to dimerize in the nucleus, possibly cotranscriptionally and aided by the Gag polyprotein [35]. This is controversial, however, since Baluyot et al. [36] examined nuclear localization of a wide range of retroviral Gag proteins and only saw avian Gag to be robustly nuclear.

The Parent lab has previously observed that trafficking of the Rous sarcoma virus (RSV) Gag protein through the nucleus is necessary for efficient genomic RNA packaging [37]. Further, they have observed interactions between RSV Gag proteins and viral RNA in the nucleus beginning shortly after transcription and continuing through export to the cytoplasm and transit to the plasma membrane for packaging [38]. Thus, this interaction appears to facilitate export and packaging of the viral RNA. RSV Gag export from the nucleus is dependent on CRM1. In this Special Issue, they show that association of RSV dimers occurs cotranscriptionally [5]. They saw dimers in the nucleus and the cytoplasm, but fewer at the plasma membrane. This would lead to the prediction that homodimers would be expected to predominate in virions, which was confirmed [5].

## 6. RNA Modifications

All retroviral RNAs are transcribed by cellular RNA polymerase II and capped cotranscriptionally by the cellular capping enzymes. Interestingly, some HIV-1 RNAs have been observed to have a TMG cap, probably due to their interactions with Rev, TGS1 (methyl transferase), and CRM1 [3,39]. Unlike normally capped mRNAs, these RNAs can be translated efficiently in the presence of inhibitors such as rapamycin [3]. Cellular RNAs exported by the CRM1-mediated route also have TMG caps (snRNAs, snoRNAs, telomerase RNA, and selenoprotein mRNAs) [3]. It will be interesting to determine whether both HIV-1 genomic and Rev-dependent mRNAs have TMG caps. In contrast, the multiply spliced HIV-1 mRNAs (Tat, Rev, and Nef) have a normal m^7^G cap, presumably because they are not exported by the CRM1 pathway. Other viral RNAs exported by CRM1 would be predicted to have TMG caps.

In addition to cap modifications, retroviral RNAs are also modified internally, much like cellular mRNAs. RSV genomic RNA was first shown in 1977 to have an average of 12 m6A residues with a consensus sequence of RGm^6^ACU [40,41,42]. Occupancy of these sites is heterogeneous; usually less than 50% methylation was observed at a specific site.

HIV-1 RNA is also methylated to generate numerous m^6^A [43,44,45] and m^5^C [46] residues. Different labs, using different methods, found differences in sites mapped as methylated and in their functional consequences. m^6^A of HIV RNA enhanced Gag protein expression and viral replication [44,45]. In addition, m^6^A at two sites in the RRE was shown to enhance its interaction with Rev [43]. The addition of m^5^C to HIV-1 RNA enhanced ribosome binding and translation [46]. Interestingly, more m^6^A residues were observed in virion RNA than in cellular HIV-1 RNA. Greatly increased (>10 fold) modification at m^5^C and 2′O-Me sites was also seen in genomic RNA in comparison to total cellular mRNA [46]. Importantly, the 2′O-Me modification, and m^6^A, have been shown to inhibit innate immune responses to HIV-1 RNA [47,48].

Overexpression of m^6^A binding proteins (YTHDF proteins) also decreased HIV-1 reverse transcription [45]. Recently, the Simon lab observed that the m^6^A reader protein YTHDF3 is packaged into HIV particles and acts as a restriction factor, reducing infectivity at the level of reverse transcription. However, the packaged HIV-1 protease cleaves this reader protein, increasing infectivity of HIV [49].

## 7. Splicing

Most cellular genes have multiple large introns that are removed during splicing, and small exons that encode proteins. However, viruses have to be more economical with their genetic material and cannot afford to discard their introns. Thus, retroviral introns are also coding sequences. In addition, retroviruses must retain a large portion of their unspliced RNA for mRNA for Gag and Pol polyproteins, as well as genomic RNA. Thus, splicing is incomplete. Over-splicing prevents viral replication [1,50]. A general principle is that the retroviral 5′ splice sites (donors) are strong and the 3′ splice sites (acceptors) are weak, facilitating incomplete and alternative splicing [1,50,51].

Like most cellular mRNAs, retroviral RNAs are spliced to generate subgenomic mRNAs, which are exported to the cytoplasm and translated by cellular ribosomes. The simplest retroviruses, such as avian leukosis virus (ALV), have a single spliced mRNA encoding the Env glycoproteins on the surface of the virion. The Env polyprotein is cleaved to generate SU (surface) and TM (trans membrane) proteins; these interact with cellular receptors to facilitate entry.

The avian retroviruses (ALV and RSV) have a negative regulator of splicing (NRS) sequence within the *gag* gene, located about 300 nts downstream from the 5′ splice site. The NRS acts as a decoy 5′ splice site that forms a nonfunctional spliceosome, thus inhibiting splicing [52]. The NRS interacts with U1 snRNP and also U11 snRNP, which is involved in splicing by minor (AT-AC) introns [53]. These binding sites overlap, suggesting U11 binding may interfere with the formation of the pseudospliceosome. Competition for U1 binding between the 5′ SS and the NRS is thought to lead to an appropriate balance of unspliced and spliced viral RNAs. Interestingly, mutations in the NRS have been associated with increased lymphoma formation in chickens, resulting from readthrough of the viral poly(A) site and splicing into downstream oncogenes, including *myb* [54,55].

The complex retroviruses, such as HIV-1, generate additional viral proteins by alternative splicing [1,16]. However, Gag and Pol proteins are still translated from unspliced RNA and Env from singly spliced mRNA. Over 50 different HIV-1 mRNAs have been described, all derived from a single full-length primary transcript (with 2 or 3 Gs at the 5′ end [28,29,30]). These fall into three size classes: full-length (10 kb), singly spliced (4 kb), and completely (or multiply) spliced (1 kb). HIV-1 Env, Vif, and Vpu proteins are translated from singly spliced mRNAs, and Tat, Rev, and Nef from multiply spliced mRNAs. The major 5′ splice site (D1) is used in all spliced HIV-1 mRNAs. The choice of 3′ splice sites and additional 5′ splice sites is modulated by the binding of cellular proteins to splicing enhancers and silencers. Rev (and possibly other export factors) plays a role in exporting mRNAs before they are completely spliced. Secondary structure is also likely to be important in alternative splicing choices. Emery and Swanstrom discuss the complex splicing regulation by HIV-1 in this Special Issue. They stress that oversplicing prevents viral replication and that it could be an antiviral strategy [1].

HIV-1 latency is regulated at both transcriptional and post-transcriptional levels, as discussed in this Special Issue by Pasternak and Berkhout [2]. There appears to be a block to splicing in latently infected cells, leading to an elevated ratio of unspliced/multiply spliced HIV-1 mRNAs [56]. This could contribute to the difficulty in reversing latency. Furthermore, unspliced viral RNA can induce an immune response without protein expression. The unspliced/multiply spliced RNA ratio is thought to be an important biomarker [2].

## 8. Polyadenylation

Retroviral RNA is polyadenylated, but the process is not efficient, and readthrough of about 15% of the ALV transcripts into downstream cellular genes is observed [57]. Mutation of the RSV NRS sequence that downregulates splicing leads to even higher readthrough of the poly(A) site [58,59] and can lead to more rapid lymphoma formation in chickens [55]. This readthrough was a major problem with the early use of retroviral vectors in human gene therapy, which can readthrough into and overexpress downstream cellular oncogenes. Readthrough of the poly(A) site also allows retroviruses to transduce cellular genes, including oncogenes.

## 9. RNA Export from the Nucleus

Unlike most cellular mRNAs, which are completely spliced before they are exported from the nucleus, retroviruses need to export unspliced, partially spliced, and completely spliced mRNAs. To solve this problem, HIV-1 encodes a regulatory protein, Rev, that binds the Rev Response Element (RRE) in *env* and mediates export through the cellular CRM1 pathway (reviewed in [60]). Rev has also been shown to play a role in promoting HIV-1 RNA translation [61]. Completely spliced HIV-1 mRNAs (Tat, Rev, and Nef) are exported by the NXF1(TAP)/NXT1 pathway, just like most cellular mRNAs that are completely spliced.

Other complex retroviruses use similar export mechanisms. HTLV-1 encodes a Rex protein that binds a Rex responsive element in the *env* gene. MMTV, a betaretrovirus, encodes an export protein called Rem as part of the N-terminus of Env [25]. Although foamy viruses are complex and have accessory proteins, they lack a Rev homolog. Instead, they use a cellular shuttle protein HuR to export their incompletely spliced RNA [22].

The simple retroviruses, which lack accessory proteins, use *cis*-acting viral sequences to interact with host export proteins. A constitutive transport element (CTE) was initially observed in the 3′ end of another betaretrovirus: the Mason-Pfizer monkey virus (MPMV) [62]. The CTE sequence forms a structured hairpin and binds directly to the cellular mRNA export factors NXF1(TAP)/NXT1 [63]. Interestingly, some cellular mRNAs, including NXF1 mRNA, also have a CTE in their intron, used for mRNA export [64].

Recently, Mougel et al. [65] proposed that MLV uses both NXF1 and CRM1 nuclear export pathways for export of unspliced viral RNAs. RNAs destined to be genomic are exported by the CRM1 pathway in conjunction with Gag proteins interacting with the Psi packaging sequence of the viral RNA. In contrast, those RNAs that will be translated use the NXF1/NXT1 pathway together with several putative CTEs in the viral RNA. MLV CTEs have been identified in the 5′ LTR, the *pol* gene, and the packaging sequence [65,66,67]. Spliced MLV mRNAs are transported by the NXF1/NXT1 pathway, as expected.

RSV has two copies of a direct repeat (DR) sequence flanking the *src* gene that is important for RNA export to the cytoplasm and likely analogous to the CTE [68,69]. These DRs have a hairpin structure [70] similar to that of the MPMV CTE. ALV, the presumed progenitor of RSV, has one copy of the DR in the 3′ UTR just upstream of the LTR. NXF1 (TAP) is required for RSV RNA export, probably through an unidentified adapter protein, as the DR has not been shown to bind directly to NXF1 [71]. Gag and CRM1 are not needed for DR-dependent export. The DR sequence has also been shown to be important for the packaging of viral RNA [72].

RSV Gag proteins shuttle between the nucleus and the cytoplasm. Leptomycin B (an inhibitor of CRM1-mediated export) inhibits this shuttling, and Gag is retained in the nucleus. There is evidence that Gag may interact with RSV full-length RNA in the nucleus and export it for packaging in a CRM1-dependent manner [37]. Thus, it is possible that RSV may also use two alternative export pathways. Early transcripts may be exported through the DR/NXF1 pathway for translation of viral mRNAs. Once Gag proteins have been synthesized and trafficked to the nucleus, genomic RNA may be exported via the Gag/CRM1 pathway for packaging. However, this is not supported by the finding that the DR sequence is important for genome packaging [72].

## 10. Translation

All retroviruses express the 5′ terminal Gag polyprotein from an unspliced primary RNA transcript. The avian retroviruses include protease at the 3′ end of the *gag* gene, while other retroviruses have protease as part of *pol.* Three upstream open reading frames have been identified in the avian retroviruses, which appear to promote virus production [73]. The Hackett lab has observed competition for full-length viral RNA between the Gag polyprotein and ribosomes, thus leading to sorting between packaging and translation [73].

Moloney MLV encodes Gag proteins with two different N-termini. Glycogag synthesis is initiated at a CUG that is 264 nt upstream of the AUG that initiates normal Gag protein synthesis. Glycogag promotes viral core stability and protects the core from APOBEC3 and other cytosolic sensors of viral nucleic acid during reverse transcription [74]. It also enhances MLV entry by antagonizing the Serinc5 protein and can substitute for HIV-1 Nef [75,76].

The next gene downstream of *gag* is *pol* in all retroviruses. *pol* is also translated as a polyprotein. In most retroviruses (with the exception of the foamy viruses) a Gag–Pol readthrough product is synthesized. In most orthoretroviruses, including RSV and HIV, a (−1) frameshift shifts the reading frame from *gag* to *pol* to generate a Gag–Pol polyprotein. A slippery sequence just upstream and a structured pseudoknot sequence downstream of the termination codon are involved in the frame shift [77]. Some retroviruses, such as MMTV and HTLV-1, have two frame shifts: one into *pro* and a second into *pol*. 

In contrast, murine leukemia viruses do not frameshift but can read through the *gag* termination codon to generate an in-frame Gag–Pol polyprotein. The MLV *gag*–*pol* readthrough may be the precursor of the frameshift at *gag*–*pol* in other orthoretroviruses; it also involves a slippery sequence upstream and pseudoknot downstream of the *gag* termination codon [78]. The foamy retroviruses encode *pol* from a singly spliced mRNA, so they do not make a Gag–Pol protein but instead a Pol polyprotein [21].

All of these readthrough or frameshift events are relatively rare (5%) so that much less Pol is translated than Gag. This promotes appropriate stoichiometry since many more capsid structural proteins are needed in a virion than enzymes. In the case of two frameshifts in MMTV, each occurs about 25% or less of the time.

## 11. Nonsense-Mediated RNA Decay (NMD)

Premature termination codons (PTCs) in the *gag* gene of RSV were seen in 1988 to be associated with degradation of viral unspliced RNA [79]. Surprisingly, PTCs near the *gag* termination codon were resistant to degradation, while those >100 nt upstream were not [80]. This RNA degradation required UPF1, suggesting it was occurring through the nonsense-mediated RNA decay (NMD) pathway [81].

Further, a sequence immediately downstream of the normal *gag* termination codon was necessary for stable RNA [82,83]. This sequence has multiple CU-rich sequences and could be truncated at both ends to generate a 155 nt minimal RSV stability element (RSE) [84]. Insertion of the RSE downstream of a PTC in *gag* led to stabilization of the RNA, supporting the hypothesis that this sequence could stabilize a termination codon that would otherwise lead to degradation [83]. The structures of all retroviral unspliced mRNAs have a very long 3′ UTR. Most of the time translation terminates after the *gag* gene, generating a 4–7 kb 3′ UTR. Even if readthrough into *pol* occurs, the 3′ UTR is still about 3 kb. In contrast, most cellular 3′ UTRs are less than 1 kb in length. In fact, longer 3′ UTRs have been associated with NMD [85].

The RSE from RSV was shown to stabilize an unstable human globin construct with a SMG 3′ UTR in human cell lines [86]. Further, the RSE was found to bind PTBP1 and block binding of UPF1 to the mRNA [86]. Several hundred human cellular mRNAs have similar CU-rich stability elements downstream of their termination codons. Further, these mRNAs tended to have longer than average 3′ UTR sequences [86]. Thus, RSV has been using a previously undiscovered cellular mechanism to stabilize its unspliced RNA. This work has clarified the mechanism of NMD that is independent of splicing and does not have a downstream exon junction complex (EJC) [85].

While it is thought that retroviral RNAs are either packaged or translated, there is evidence that RSV RNA may be translated and then packaged. When premature termination codons were inserted into the RSV *gag* gene, the RNA underwent NMD and was present at about 10% of normal levels. Interestingly, the genomic RNA in virions was also present at a reduced level, suggesting it underwent NMD before packaging. Since NMD is coupled to translation, this suggests that the packaged RNA was previously translated [87].

Another mechanism is used to evade NMD by HTLV-1 [88]. Both viral Tax and Rex proteins have been found to have the ability to inhibit NMD. Tax bound UPF1 and the translation initiation complex component INT6/eIF3E, resulting in partial inhibition of NMD and the stabilization viral RNA [89]. Rex was even more efficient than Tax in inhibiting NMD [90].

It is not clear how other retroviruses manage to evade NMD, but they all form aberrant unspliced mRNAs with long 3′ UTRs. One possibility is that the TMG cap of HIV-1 RNA exported with Rev/CRM1 may also inhibit NMD, although this has not been investigated. NMD involves interactions with the CBP80/CBP20 nuclear cap-binding complex [91], and the TMG capped HIV-1 RNA binds instead to CBP80/NCBP3 [3].

## 12. Closing

Studies of retroviral RNA processing have taught us a lot about cellular RNA processing, especially about RNA nuclear export [60,63], specialized translation and frame-shifting [3,77], and protection from NMD [82,83,86]. In addition, it seems that RNA structure, particularly of the 5′ UTR, is important in sorting RNA between packaging and translation. It is likely also important in other areas of retroviral and cellular RNA processing, including alternative splicing, translation, and RNA modification.

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
