# Peer review of "Retroviral RNA Processing"

_viruses, 2022, doi:10.3390/v14051113_

Round 1

Reviewer 1 Report

This review covers the topics in the special issue on RNA processing and develops the larger context of RNA processing in retroviruses. It brings together a vast amount of information in a cogent, fluid manner. There is some redundancy in the text - between the overview sections and the later, more granular sections, but this does not detract from the quality of the document. A few relatively minor revisions are requested to clarify a few statements and points. The document is an appropriate overview of the articles in the special issue, and stands alone as a valuable review on the scope and types of RNA processing events regulating retroviruses.  

line 56

"...major mRNA that generates Gag and Gag-Pol"

As written, the sentence insinuates there are other mRNAs that engender these structural proteins. Please revise to clarify this point.

line 67

Is there a study data to indicate this or rough call? :

'at least half of the primary transcripts must remain unspliced'

Please clarify this statement.

line 109

confusing sentences, please revise. ie Pol encodes protease and RT, which are expressed as a fusion protein and integrase is....

line 110

does this mean:

"to facilitate packaging, RNA binding activity is attributed to Pol.

Please clarify

line 116

typo, TaS, not tat

line 381

add

Specialized translation [3, PMID: 34949712]

Ref 46 is redundant with 42

Author Response

I thank the reviewer for the positive comments and the helpful suggestions.

  1. Line 56:  Major was deleted
  2. Line 67:  I have softened the statement and added 2 references.  The numbers came from "Principles of Virology" for RSV but I don't want to cite that.
  3. Line 109:  rewritten for clarity.
  4. Line 110:  PFV Pol interacts with Gag and possibly also RNA.  Rewrote sentence and added reference 22.
  5. Line 116:  typo corrected
  6. Line 381:  Specialized translation added
  7. Redundant reference deleted-sharp eyes.

Reviewer 2 Report

This summary review is well done and highly informative. However, in my opinion there are a few points where it could be improved.

Line 37: the text gives the impression that the protease is within the viral capsid. I do not believe that the protease is within the capsid of the mature particle. Perhaps the author meant that it is within the particle. In any case the statement is confusing or misleading and should be corrected.

Line 60 lists the products of the gag gene as matrix, capsid, and nucleocapsid. This statement sounds as if this is a complete list, whereas, of course, most retroviruses synthesize one or more additional proteins in Gag. It should be clarified.

Line 105 states that there is no frameshifting or suppression of gag termination. This is not self-explanatory and thus is mystifying. The same is true of the statement (line 110) that “there is no Gag-Pol polyprotein to facilitate packaging Pol.” Both of these are in contrast to orthoretroviruses (which should be made explicit) but should be explained for the benefit of the reader who is not fully familiar with the subject (while a fully knowledgeable reader would not need this review in any case).

Line 139: the heterogeneity of start sites in HIV RNAs and the preferential packaging of one of these species was first reported by Masuda et al. (Scientific Reports, 2015) and these authors should certainly be cited as well as Kharytonchyk et al. and Brown et al.

Line 158 seems to suggest that MLV Gag enters the nucleus, since it is said to promote dimerization and dimerization is said to occur in the nucleus. However, I know of no evidence that MLV Gag is ever nuclear, and I would urge the addition of a little healthy skepticism to this discussion. The claim that the RNA dimerizes in the nucleus is based on experiments with some highly unnatural genomic constructs (ref. 30).

On the other hand, it is well-documented that RSV Gag enters the nucleus. The review states (line 166) that interaction between RSV Gag and RSV RNA appears to facilitate export and packaging of the RNA. Perhaps it should also mention that an RSV with a modified Gag has virtually full infectivity but does not seem to enter the nucleus, at least not with a CRM1-dependent export pathway, casting some doubt on the idea that nuclear cycling of RSV Gag is essential for virus replication (Baluyot et al., JV 2012).

Line 179 states that TMG-capped HIV RNAs are translated efficiently even when mTOR is inhibited. Again, this is not self-explanatory: many readers will not have any idea why mTOR is mentioned here. The statement is repeated in line 334.

Line 288 mentions MLV CTEs. It should cite Bartels & Luban, Retrovirology 2014, and Pilkington et al., Nucleic Acids Research 2014, as well as ref. 64.

Line 298 mentions “LMB”; it would be good to define this acronym.

Line 315 describes the function of MLV Glyco-Gag as promoting core stability and protecting from APOBEC3. This is one view of its function, but there is certainly more to the story. It is well documented that it protects the virus against the effects of Serinc5, as documented in Pizzato, PNAS 2010; Usami et al., Nature 2015; Rosa et al., Nature 2015; and Ahi et al., mBio 2016.

The paragraph beginning at line 334 is completely incomprehensible. It leaves too much to the imagination of the reader.

Author Response

I thank the reviewer for the helpful comments and additional references.  I have incorporated the changes in the manuscript.

Line 37: the text gives the impression that the protease is within the viral capsid. I do not believe that the protease is within the capsid of the mature particle. Perhaps the author meant that it is within the particle. In any case the statement is confusing or misleading and should be corrected.

      Protease was deleted from line 37.  Thank you for pointing that out.

Line 60 lists the products of the gag gene as matrix, capsid, and nucleocapsid. This statement sounds as if this is a complete list, whereas, of course, most retroviruses synthesize one or more additional proteins in Gag. It should be clarified.

     This sentence was modified.

Line 105 states that there is no frameshifting or suppression of gag termination. This is not self-explanatory and thus is mystifying. The same is true of the statement (line 110) that “there is no Gag-Pol polyprotein to facilitate packaging Pol.” Both of these are in contrast to orthoretroviruses (which should be made explicit) but should be explained for the benefit of the reader who is not fully familiar with the subject (while a fully knowledgeable reader would not need this review in any case).

     This section was rewritten to clarify.

Line 139: the heterogeneity of start sites in HIV RNAs and the preferential packaging of one of these species was first reported by Masuda et al. (Scientific Reports, 2015) and these authors should certainly be cited as well as Kharytonchyk et al. and Brown et al.

     Reference was added.

Line 158 seems to suggest that MLV Gag enters the nucleus, since it is said to promote dimerization and dimerization is said to occur in the nucleus. However, I know of no evidence that MLV Gag is ever nuclear, and I would urge the addition of a little healthy skepticism to this discussion. The claim that the RNA dimerizes in the nucleus is based on experiments with some highly unnatural genomic constructs (ref. 30).  

     This section was rewritten.

On the other hand, it is well-documented that RSV Gag enters the nucleus. The review states (line 166) that interaction between RSV Gag and RSV RNA appears to facilitate export and packaging of the RNA. Perhaps it should also mention that an RSV with a modified Gag has virtually full infectivity but does not seem to enter the nucleus, at least not with a CRM1-dependent export pathway, casting some doubt on the idea that nuclear cycling of RSV Gag is essential for virus replication (Baluyot et al., JV 2012).

     The Baluyot reference was added to support RSV Gag in the nucleus but not the other Gags.  The modified Gag experiment with RSV seemed artifcial and was not included.

Line 179 states that TMG-capped HIV RNAs are translated efficiently even when mTOR is inhibited. Again, this is not self-explanatory: many readers will not have any idea why mTOR is mentioned here. The statement is repeated in line 334.

     This was rewritten.

Line 288 mentions MLV CTEs. It should cite Bartels & Luban, Retrovirology 2014, and Pilkington et al., Nucleic Acids Research 2014, as well as ref. 64.

     Thank you for those references.

Line 298 mentions “LMB”; it would be good to define this acronym.

     Done

Line 315 describes the function of MLV Glyco-Gag as promoting core stability and protecting from APOBEC3. This is one view of its function, but there is certainly more to the story. It is well documented that it protects the virus against the effects of Serinc5, as documented in Pizzato, PNAS 2010; Usami et al., Nature 2015; Rosa et al., Nature 2015; and Ahi et al., mBio 2016.

     This additional function was added and two of the references were added.

The paragraph beginning at line 334 is completely incomprehensible. It leaves too much to the imagination of the reader.

     This paragraph was deleted.  Two of the sentences were used elsewhere.